# Microbial Community Structure in Ancient European Arctic Peatlands

**DOI:** 10.3390/plants11202704

**Published:** 2022-10-13

**Authors:** Alexander Pastukhov, Vera Kovaleva, Dmitry Kaverin

**Affiliations:** 1Institute of Biology Komi Science Centre Ural Branch Russian Academy of Sciences, Kommunisticheskaya 28, 167982 Syktyvkar, Russia; 2BIO-GEO-CLIM Laboratory, National Research Tomsk State University, Lenina 36, 634050 Tomsk, Russia

**Keywords:** amino acids, bacteria, archaea, fungi, cryolithozone, bogs, peat plateaus, nitrogen

## Abstract

Northern peatlands, which are crucial reservoirs of carbon and nitrogen (415 ± 150 and 10 ± 7 Pg, respectively), are vulnerable to microbial mineralization after permafrost thaw. This study was carried out in four key sites containing northern permafrost peatland, which are located along the southern cryolithozone. The aim of this study is to characterize amino acids and the microbial community composition in peat strata along a climate gradient. Amino acids and microbiota diversity were studied by liquid chromatography and a quantitative polymerase chain reaction. The share of amino acid fragments was 2.6–7.8, and it is highly significantly correlated (r = 0.87, −0.74 and 0.67, *p* ˂ 0.05) with the organic nitrogen concentration in the soil, the C/N ratio, and *δ*^15^N. The data shows the existence of a large pool of microorganisms concentrated in permafrost peatlands, and a vertical continuum of bacteria, archaea, and microscopic fungi along the peat profile, due to the presence of microorganisms in each layer, throughout all the peat strata. There is no significant correlation between microorganism distribution and the plant macrofossil composition of the peat strata. Determining factors for the development of microorganism abundance are aeration and hydrothermal conditions. The availability of nitrogen will limit the ability of plants and microorganisms to respond to changing environmental conditions; however, with the increased decomposition of organic matter, amino acids will be released as organic sources of nitrogen stored in the protein material of peat-forming plants and microbial communities, which can also affect the organic nitrogen cycle.

## 1. Introduction

The northern peatlands, which occupy 3.7 ± 0.5 million km^2^, contain 415 ± 150 Pg of carbon and 10 ± 7 Pg of nitrogen; this accounts for approximately half of the area, and the reserves of permafrost are affected [1]. These systems provide a global carbon sink for the atmosphere as CO_2_ and carbon accumulates in the form of peat deposits; however, peatland systems also act as powerful sources of greenhouse gases (methane and nitrogen dioxide emissions) due to climate change and permafrost degradation [2]. The accumulation of organic carbon in the peatlands occurs as a result of the predominance of organic matter being inputted over the rate of decomposition, which is directly related to the activity of microorganisms. Microbial communities not only determine the processes that establish the transformation of organic matter, but they are also important for mineralization and nutrient absorption by plants, and they affect the productivity and functioning of ecosystems as a whole [3]; therefore a better understanding of the microbial community is needed to apply to the extreme conditions of northern peatlands. Despite the development of new methods and techniques in microbiology, which have contributed to obtaining a large amount of data on the diversity and composition of microbial communities, the mechanisms concerning the functioning and sharing of various microorganism groups during the processes that cause the transformation of organic substances in the northern peatlands still remain unclear. It is likely that these processes are carried out by different mutually complementary groups of microorganisms.

We need a better, in-depth knowledge of microorganisms, in order to better understand microbial activities in a specific peatland, and across different peatlands. This study of microbial communities in peatlands was focused on researching individual functional groups [4,5,6,7,8]. The study of microbial communities in peat soils was carried out on a relatively small scale, limited to a small number of samples, often without taking into account spatial and temporal variations [9,10,11,12]. In particular, in the anaerobic conditions of the deep peat layers (cathotelm), microbial communities have been not taken into account, and the microbiological carbon and nitrogen cycling features in permafrost peat strata have not been studied [13,14,15]; however, to explain the processes of peat decomposition and nutrient cycling, to assess future changes in the global carbon and nitrogen balance, and to develop scenarios for the peatlands’ evolution in the cryolithozone, we need information concerning the composition and structure of microbial communities, and the influence of these indicators on the processes of organic matter transformation.

The processes that determine the microbiological decomposition of peat are associated with a combination of factors that limit its destruction (acidic reactions in the environment, low temperatures, low oxygen concentrations, and lack of nutrients). The ecological conditions of the peatlands often limit the availability of nitrogen nutrition for microorganisms; therefore, amino acids (AAs) are an important source of nitrogen. As AAs are water-soluble forms of bioavailable organic nitrogen, they are directly absorbed by microorganisms and plants in the soil [16,17,18,19]. There is a relationship between microorganisms and amino acid content; AAs contribute to the activation of microbiological processes, and moreover, a possible method for synthesizing free amino acids using microorganisms, which can be released and accumulated in the soil, has been revealed [20]. In addition, the more microorganisms in the soil, the higher the amino acid content [21].

The amino acid availability for absorption by microorganisms will depend on climate change, which may affect changes in the carbon and nitrogen cycles of the peatlands during permafrost degradation. Seasonal changes in AA availability may limit or increase nitrogen availability, and thus AAs can affect vascular plant growth and the rate of degradation of peatland organic matter as a result of microorganisms. Rising temperatures may contribute to the degradation of permafrost in permafrost peatlands, as well as lower groundwater levels in non-permafrost bogs, thus accelerating CO_2_ and NO_2_ emissions due to the decomposition of large stocks of carbon and nitrogen [22,23].

The composition heterogeneity and peat organic matter properties in the acrotelm and catotelm of peat strata were shown in our previous studies [24,25,26], which denoted the various paleogeographic conditions throughout their evolution [27,28,29]. The purpose of this study is to fill the gaps in our previous research now that structural features have been determined and an assessment of the profile distribution of microorganisms has been undertaken, the latter of which found correlations with the amino acid content in the northern permafrost peatlands at the southern cryolithozone limit.

## 2. Results and Discussion

### 2.1. Soil Properties

The Eastern European permafrost peatlands have a more acidic pH (pH_H2O_ = 4.5 ± 0.5 and pH_KCl_ = 3.6 ± 0.4) in comparison with the pH levels found in the Western Siberian peat plateaus (4.7 ± 0.5 and 3.7 ± 0.4, respectively) [29]. The pH values thus increase down the peat strata. The mean organic carbon content is 48.5 ± 1.8, and the minimum values can be explained by the predominance of weakly decomposed wood residues in the peat (Figure 1). Nitrogen concentrations form a wider range, from 0.8 to 4.9, with an average value of 2.5 ± 0.5. Peat deposits have minimal ash content, at 94.7 ± 1.8, except for individual horizons which have a large amount of weakly decomposed wood and moss remnants. Moreover, in the Eastern European peatlands, the ash content gradually increases, whereas in Western Siberia, its values are rather heterogenic throughout the profile.

The C/N ratios indicate that the degree of peat decomposition, and the level of nitrogen enrichment in peat, are very different in the peatland profiles, varying from 12 to 75, with an average value of 25 ± 5.7. High C/N ratios are usually in near-surface layers (0–20 cm), unless peat circles (peat plateau sites with bare patches) (Kolva 1) and/or eroded upper peat horizons exist (Inta 11). In the lower active layers, and the so-called transitional layers (45–75 cm), which could thaw in the warmest years, the C/N ratios are minimal (12–31). This is apparently due to their maximum moisture saturation, and as a result, increased microbiological activity occurs, as well as a change in the plant macrofosssil composition of most peat deposits; usually, this means that slightly decomposed shrub-moss peat is replaced by well-decomposed grass, grass-moss, or grass-horsetail moss peat. The C/N ratio in the permafrost gradually increases down the peat strata, with the exception of layers that have numerous weakly decomposed shrubs and tree residues.

The atomic ratio H/C (1.27 ± 0.06) in both soils indicates the predominance of aliphatic chains in HA molecules which are in the presence of 40–50 aromatic structures. The values of the oxidation degree (O/C) are 0.49 ± 0.06, thus indicating that the oxidation process only dominates in the active layer.

Differences in physical and chemical properties show changes and inequalities in paleogeographic conditions during the formation of the permafrost peatlands. The chemical properties of the peatlands depend on the composition and structure of plant communities and their changes during the peatlands’ evolution, as well as on the conditions of soil nutrition.

### 2.2. Amino Acid Composition

AAs are one of the carbon and nitrogen sources for bog ecosystems. The ecological wetland conditions often limit the availability of nitrogen, and slow decomposition leads to the accumulation of large carbon stores. Climate change may accelerate carbon being released from the peatlands, which can be affected by limited nitrogen availability. The available AAs for uptake by microbes and plants are also likely to be influenced by climate warming and can affect the peatland carbon cycle.

Seventeen different amino acids have been identified in the organic matter of the studied peatlands. The mass fraction of AAs in dry peat varies from 2.6 ± 0.4 to 7.8 ± 1.3 (from 24.8 to 40.6 of the soil’s organic nitrogen concentration) (Figure 2; Appendix A). The sum of AA in peat horizons significantly correlates not only with the soil’s organic nitrogen content and *δ*^15^N isotopes (r = 0.87 and 0.67, respectively, *p* ˂ 0.05), but also with the C/N ratio (r = −0.74, *p* ˂ 0.05), which indicates the degree of decomposition for organic matter. The AA accumulation is facilitated by low temperatures, acidic reactions in the environment, and reduced microbiological activity in the peatlands; that is, factors that determine the humification degree of the soil’s organic matter.

A rather significant spread in the values concerning AA concentration is explained by the differences in the plant macrofossil composition of peat strata. Indeed, in the above-ground part of the shrub and herbaceous vegetation, a more significant nitrogen accumulation occurs in contrast to mosses and lichens. Second, the low reserve of AA fragments in the peatlands is defined by cyclical biological characteristics in the soil–plant system. Nutrients entering the soil are intercepted by living roots during the slow decomposition of dwarf shrub remnants and moss–lichen litter [30]. The variability of physicochemical properties and AA concentrations in peat indicate that there were differences in the paleogeographic conditions of the peatland during its genesis, as well as differences in terms of its ground water nutrition; thus, the lowland peat plateau (site Inta 1), during its initial stages, was mainly vegetated by eutrophic herbaceous–hypnum communities with a predominance of hygrophilic species, which were replaced by mesotrophic shrubs, and later replaced again by hygrophilic eutrophic plants. In foothill peat plateaus (sites Inta 11 and Voda-Ty), mesoeutrophic herbaceous communities prevailed at all stages of its genesis [24,26], which is reflected in the greater enrichment of organic matter with nitrogen.

The content of individual AAs in the organic matter composition of peatlands can vary significantly both throughout all peat strata, and between individual peat plateaus; however, the general patterns of AA accumulation fit into a single amino acid spectrum (Appendix A). In accordance with previous studies, in general, the uniformity of the AA composition is typical for humic substances (humic and fulvic acids), microorganisms, plants, and soils [31].

The structure of organic matter in peatlands can be described with five AAs: asparagine, glutamine, leucine, alanine, and glycine, the total amount of which varies from 47.0 to 53.1. The first four primary AAs are generated by direct and reductive amination. These AAs are always contained in plant protein, as they synthesize the primary binding of ammonia [32]. The composition of organic matter is dominated by neutral AAs (63.6–71.4%), among which, the relative molar fraction of aromatic AAs is 5.8–10.1, and of hydroxy AAs, it is 11.2–16.1; hence, neutral AAs are the most stable amino acids [33]. The acidic AA share is 18.5–22.1, and the proportion of basic AAs varies from 8.5 to 12.0. The share of diamino acids ranges from 7.5 to −10.2, and for heterocyclic AAs, it is 4.4–8.8 (Appendix A). The AA composition of organic matter differs both in absolute quantity, in relative AA group content, and it contains a higher yield of acidic AAs than basic AAs, which is due to the high acidity of the peat.

The biological soil activity can be illustrated using the molar fraction ratio of hydroxy and heterocyclic AAs, and this ratio can be a humification depth marker. A significant correlation is found between the molar concentration of this coefficient and the values of the *δ*^15^N, the soil’s organic nitrogen, and the C/N ratio were r = 0.66, 0.51, and −0.47, n = 42, *p* < 0.05, respectively (Figure 2).

Our data shows a similar quantitative neutral and dicarboxylic AA composition for the studied peatlands. Regarding the composition of organic matter, the share of cyclic AAs is low (14.5 ± 0.9), but its range of values is higher (11.5–17.0), which obviously indicates a noticeable differentiation between the molecular compositions of peat. Based on the data analysis concerning the AA composition of organic matter, it can be concluded that the general patterns of AA formation in the permafrost peatlands have the same character as the AA spectrum. Moreover, they also have a degree of individuality in terms of changing the quantitative ratios of the main AA groups, and the absolute values of AAs, in the organic matter of peatlands at the various stages of their geneses.

### 2.3. The Microbial Community Diversity and the Distribution of Microbial Groups in Peat Strata

Cryic histosols in peat plateaus have low pH values, temperatures, oxygen concentrations under water saturated conditions, and a high content of phenolic compounds [34]. The organic matter destruction of *Sphagnum* leads to phenol-containing compound formation, which contributes to the conservation of the organic matter of plants in the peatlands [35]. Since the processes of organic matter decomposition are very complex and have multiple stages, they are carried out by many microbial groups, concentrated in a peat stratum. Moreover, since the vertical stratification of microorganisms is associated with organic matter distribution, microorganisms are present in the peatlands throughout the peat strata [36]. Furthermore, in deep peat layers, microorganisms are in a viable state, as evidenced by their growth on nutritious media [35,37].

#### 2.3.1. The Bacterial Community Distribution

Bacteria were identified in all layers of the studied peatlands. The number of copies of bacterial ribosomal genes in the studied peat samples varies widely—from 5.55 × 10^7^ ± 6.38 × 10^6^ to 1.94 × 10^10^ ± 1.61 × 10^9^ copies/g of soil—depending on the peatland and sampling depth (Figure 3).

At site Inta 1, the maximum numbers of bacteria were recorded in the upper peat layers (Figure 3a). The maximum number of bacterial gene copies is 1.16 × 10^10^ ± 3.43 × 10^8^ copies/g of soil at a depth of 0–10 cm, and the minimum number is 1.34 × 10^8^ ± 9.23 × 10^5^ copies/g of soil at a depth of 90–100 cm. In the lower peat layers, there is no sharp decrease in the density of bacterial groups along the profile, but there is a gradual decrease in number.

At site Kolva, the profile distribution of the number of bacteria has a wave-like character, with maximum values of 1.22 × 10^10^ copies/g of soil at a depth of 100–110 cm (Figure 3b). The minimum value found at a 60–70 cm depth was 2 × 10^8^ copies/g of soil.

The analysis of the obtained data showed a similarity between sites Inta 11 and Inta 11; this is due to the fact that the abundance of bacteria decreases down the profile (Figure 3c). The maximum number of bacterial genes is 1.94 × 10^10^ copies/g of soil at a depth of 0–10 cm, which decreases down the profile, and reaches the minimum values of 1.91 × 10^9^ copies/g of soil at a depth of 40–50 cm. The obtained data concerning the number of copies of bacterial genes in soils, from sites Inta 11 and Voda-Ty, complement each other. At site Voda-Ty, the number of bacteria is generally lower than at site Inta 11; indeed, the maximum number of copies of bacterial genes is 8.7 × 10^9^ copies/g of soil at a depth of 190–210 cm, and the minimum number is 5.6 × 10^7^ copies/g of soil, found at a depth of 160–170 cm (Figure 3d).

Bacteria comprise the largest group of microorganisms in the peatlands. The abundance of this group remains relatively high throughout the peat strata. Bacteria determine the direction and pace of biochemical processes in peatlands.

#### 2.3.2. The Archaeal Community Distribution

Anaerobic conditions develop in the lower layers of peat strata, and since the metabolic capabilities of most strict anaerobic bacteria are limited, the bacterial decomposition of complex organic compounds (i.e., to CH_4_ and CO_2_) depends on cooperation with other groups of microorganisms, such as archaea, which produce methane in oxygen-free zones at a low redox potential [38,39].

In all studied samples, the number of archaeal genes varies from 3.4 × 10^6^ to 1.0 × 10^10^ copies/g of soil. Archaea, as well as bacteria, were found across all peat strata in the studied peatlands.

At site Inta 1 (Figure 4a), the maximum number of archaeal genes was observed in the upper peat layers, with the maximum number reaching 8.6 × 10^9^ copies/g of soil at a depth of 20–30 cm and the minimum number reaching 3.4 × 10^6^ copies/g of soil at a depth of 40–50 cm.

The profile distribution of archaea in sites Kolva, Inta 11, and Voda-Ty is similar to the bacterial distribution in the peatlands. At site Kolva, the number of archaeal genes decreases, reaching a minimum value of 2.9 × 10^7^ copies/g of soil at 80–90 cm, which then increases sharply by two orders of magnitude, 5.1 × 10^9^ copies/g of soil, at a depth of 100–110 cm (Figure 4b). The maximum number of archaeal 16S rRNA genes at site Inta 11 (Figure 4c) is detected in the 0–10 cm layer, amounting to 1 × 10^10^ copies/g of soil, and the minimum is at a depth of 40–50 cm, amounting to 5.3 × 10^8^ copies/g of soil. At site Voda-Ty, the minimum number of archaeal ribosomal operons is found in the 160–170 cm layer, amounting to 1.6 × 10^7^ copies/g of soil, and the maximum number is found at a depth of 190–210 cm, amounting to 7.5 × 10^9^ copies/g of soil (Figure 4d).

The profile distribution of archaea is consistent with the distribution of bacteria. The high abundance of archaea is associated with their function during the decomposition and transformation of organic matter under semi-anaerobic and anaerobic conditions.

#### 2.3.3. Fungal Distribution

There are many factors inhibiting microbial activity in the peatlands: low temperature, semi-anaerobic and anaerobic conditions, high acidity, lack of mineral nutrients, phenolic compound toxicity, substrate dissociation, enzymes, and microbial cells [40,41,42]; however, the significance of individual factors will be different for diverse groups of microorganisms. For fungi development in the peatlands, aeration is an important limiting factor, since most of them are aerobic organisms; therefore, the total number of fungi in the studied peatlands varies from 1.45 × 10^6^ ± 4.48 × 10^5^ to 1.17 × 10^10^ ± 4.40 × 10^8^ copies/g of soil, which is significantly lower than the number of bacteria and archaea.

At site Inta 1, the maximum number of micromycetes, which is 1.17 × 10^10^ ± 4.40 × 10^8^, is found at a depth of 0–10 cm, and the minimum number, which is 1.45 × 10^6^ ± 4.48 × 10^5^ copies/g of soil, is found at a depth of 60–70 cm (Figure 5a).

At site Kolva, the maximum number of fungal ribosomal operons is 7.02 × 10^8^ ± 9.61 × 10^7^ copies/g of soil, which was found in the 5–10 cm layer, and the minimum number is 8.02 × 10^7^ ± 4.11 × 10^6^ copies/g of soil, found at a depth of 30–40 cm (Figure 5b).

At site Inta 11, the number of copies of the 18S rRNA ribosomal genes of microscopic fungi varies from 9.69 × 10^9^ ± 6.02 × 10^8^ copies/g of soil at a depth of 0–10 cm, to 9.74 × 10^7^ ± 4.98 × 10^6^ copies/g of soil at a depth of 20–30 cm (Figure 5c).

At site Voda-Ty, the micromycete distribution along peat strata is similar to the bacterial and archaeal density; the minimum number, which is 2.47 × 10^7^ ± 3.48 × 10^6^ copies/g of soil, can be detected at a depth of 160–170 cm, and the maximum number, which is 2.46 × 10^9^ ± 3.28 × 10^8^ copies/g of soil, can be detected at a depth of 190–210 cm (Figure 5d).

In the studied peatlands, the number of fungi is much lower than that of bacteria and archaea. Since fungi are most sensitive to the conditions in peatlands, most of them are represented by spores, especially in the permafrost layers, and thus, they do not carry any functional load; nevertheless, they are a necessary pool for maintaining the homeostasis of the microbial community.

### 2.4. General Patterns and Correlations

The obtained results for the microorganism distribution in the permafrost peatlands show that the lower peat layers have a relatively high microbial community density. This specificity of peat soils is also typical of the peat plateaus of the Eastern European and Western Siberian Plains [43].

The vertical profile distribution of the number of bacteria, archaea, and micromycete gene copies at sites Inta 1 and Inta 11 shows a general regularity; their maximum number occurs in the upper layer, at a depth of 0–10 cm, and then it decreases by one to two orders of magnitude down the profile. The upper horizons have high biogenicity due to the processes related to organic matter transformation, which are associated with microorganism activity (mineralization of plant residues, synthesis of humic substances, biogenic accumulation of micro- and macroelements). A decrease in the amount of DNA in peat strata explains a change in the environmental conditions (a decrease in the soil’s organic carbon stocks, a change in the chemical composition and its physical characteristics).

A high abundance of microorganisms was found at sites Kolva and Voda-Ty in deep peat layers at depths of 100–110 and 190–210 cm, respectively; however, we cannot be certain that the pool of microorganisms is in a functionally active state in the lower peat layers. In the deep layers of the peatlands, hydrothermal and redox conditions, as well as the presence of permafrost, are unfavorable for microorganisms, which contributes to bacteria transitioning to a dormant state, thus leading to the conservation of a large proportion of bacterial biomass in peat [43,44].

Our study shows that in all the peatlands, fungi are commonly found throughout the entire profile of the peat strata. At the same time, at sites Inta 1 and Inta 11, the abundance of micromycete is maximal in the upper layers, and it then decreases down the profile, since fungi are active decomposers of fresh plant litter under conditions of sufficient aeration in the upper acrotelm. Site Kolva has a relatively low number of micromycetes compared with the number of bacteria and archaea, due to insufficient aeration and anaerobic conditions, as well as the minimal input of fresh organic matter, thus causing the number of fungi to decrease. At site Voda-Ty, a high number of 18S rRNA ribosomal gene copies can be found at a depth of 190–220 cm. Since fungi are more aerobic microorganisms, and have the ability to quickly respond to adverse environmental conditions in the lower peat layers, they move into anabiosis. Fungal spores pass into a state of exogenous dormancy in the catotelm. When in the physiological state of being a spore, fungi can stay dormant for a long time, keeping their viability in the multimeter peat strata and maintaining a high level of “living” carbon reserves in bog ecosystems [45].

The contours of the microorganism distribution of peat strata depend little on the plant macrofossil composition of peat. Indeed, in the Inta 11 peatland, five types of peat alternate in a layer that is over a meter thick, from grass-*Sphagnum* to *Scheuchzéria*; in the Kolva peatland, there are also five types, from shrub-*Erióphorum* to eutrophic grass communities; and in the Voda-Ty peatland, there are four types, from sedge-*Erióphorum* to willow-sedge phytocenoses; however, such an alternation of peat layers does not affect microorganism distribution in the peat strata. A correlation analysis revealed a weak inverse correlation (r = −0.2–−0.4) between the abundance of all microorganism groups and plant macrofossil composition (Figure 6).

The number of microorganisms is most significantly inverse correlated (r = −0.8, *p* ˂ 0.01) with the moisture/ice content of peat layers. As the humidity of the peat increases, aeration decreases, and the pore space is filled with water, which negatively affects the development and functioning of the microbial complex. The strong inverse predictor (r = −0.6, *p* ˂ 0.05) is the number of microorganisms that represent the total nitrogen content, which is associated with the processes of microbiological organic matter transformation in a peat stratum. Microorganisms quickly assimilate with monomeric nitrogen compounds.

The abundance of bacteria, archaea, and microscopic fungi is highly significantly correlated with the C/N ratio (r = 0.6–0.8). Numerous factors that limit the vital activity of microorganisms in permafrost peatlands lead to a slowdown in decomposition processes, and as a result, an increase in the soil’s organic carbon stocks.

## 3. Materials and Methods

### 3.1. Study Area

The research area is located within the northern treeline and southern border of the modern cryolithozone, which hosts a massive island permafrost (Figure 7). Mean annual temperatures vary from −1.5 to −3 °C, and mean annual precipitation fluctuates from 600 to 800 mm. Except in the Ural Mountains, the study area is covered by flat wetlands, with elevations from 40 to 160 m above sea level, which are underlain by Upper Neopleistocene glacial and lacustrine deposits. According to IPCC reports [46], this region is most vulnerable to climatic and/or anthropogenic changes. The key study sites were selected according to the comparative geographical principle, taking into account the current and predicted geocryological situation in the study area. Sites Voda-Ty (V) and Inta (I) are located in the extreme northern taiga, and site Kolva (K) is in the southern tundra. In the Pechora Plain, peat plateaus with thin permafrost tables (10–15 m) are interspersed with permafrost-free fens and thermokarst lakes that are open to, and/or filled in with, vegetation of the southern tundra (K—“Kolva”: K 1 and K 2—63°42′ N, 75°54′ E) and the extreme northern taiga (V—“Voda-Ty”: 65°26′ N, 60°49′ E; I—“Inta”: I 1—65°54′ N, 60°26′ E; I 11—66°05′ N, 59°58′ E).

According to the classification scheme of the mire vegetation proposed by T.K. Yurkovskaya [49], the studied peatlands are dwarf birch (*Betula nana*) grass-shrub-moss-lichen on mounds (*Cetraria nivalis*, *Dicranum elongatum*, *D. congestum*, *Polytrichum alpestre*, *Ledum palustre*, *Vaccinium uliginosum*, *V. vitis-idaea*, *Rubus chamaemorus*), and cottongrass-sedge-hypnum-*sphagnum* in fens (*Sphagnum lindbergii*, *Drepanocladus fluitans*, *Carex rariflora*, *C. rotundata*, *Eriophorum russeolum*) and large mounds in North European forest–tundra bogs.

In accordance with the WRB, the peat plateau soils are Ombric Sapric Cryic Histosols (Hyperorganic) [50]. These soils have well-decomposed organic material (peat) that is more than 2 m deep, they predominantly obtain their nutrition from the atmosphere, and they have underlying permafrost that is within 1 m. The permafrost-free fen soils are identified as Hemic Histosols. The peat circle soils (bare surfaces on peat plateaus) belong to Ombric Sapric Cryic Histosols (Hyperorganic Turbic), because upper horizons are slightly cryoturbated.

### 3.2. Soil Sampling and Laboratory Analyses

#### 3.2.1. Soil Sampling

Histosols were described and sampled on the tops of peat mounds (peat plateaus) and in the center of the adjacent fens. The peat was sampled using a 10–15 cm vertical resolution. In the active layer of peat plateaus, peat was sampled into measuring cylinders (volume 503 cm^3^) to determine the volumetric weight and weight moisture content of each sample; in the permafrost layer, frozen cores were drilled to a depth of up to 10 m by a machine tool, UKB 12/25-02 “Pombur”; and in the fens, a Russian peat corer was applied (a steel cover plate freely rotated inside the core’s tube). The diameter of the measuring cylinder and drilling machine pipe was 100 mm.

#### 3.2.2. Plant Macrofossil Analyses

Plant macrofossil analysis was determined at the Institute of Biology Karelian Science Center, Russian Academy of Sciences. The samples were detected after deflocculating peat of a known volume (5–20 cm^3^) with 5 KOH and sieving it (150 μm mesh) to remove fine detritus; plant residues were identified by a stereo binocular (25–40× magnification) using the literature as a reference [51]. Species of *Sphagnum* were examined according to their leaf morphology [52] with a microscope (100–400× magnification). The abundance of selected plant macrofossil species was calculated as volume percentage of the total macrofossil assemblage (below 1% was marked as +). Gross stratigraphy and plant macrofossils were visualized using the software “Korpi” [53].

#### 3.2.3. Element Analyses

The quantitative chemical analyses of the peat were determined at the certified Ekoanalit laboratory of the Institute of Biology Komi Science Center of the Ural Branch of the RAS. The elemental quantitative analysis of the organic compounds of carbon, hydrogen, and nitrogen was determined using an automatic CHN(S,O) analyzer EA-1110 (Carlo Erba, Italy), which is based on a combination of sample pyrolysis in oxygen in a vertical flow reactor type, followed by gas chromatographic separation of the pyrolysis products. Data were corrected for water and ash content. The oxygen share was calculated from the difference, taking the total content of C, N, H, O as 100. The ratios of the elements (C/N, H/C, and O/C) were based on mole percentages. Hygroscopic moisture was measured gravimetrically, and the ash content was estimated from loss-on-ignition (LOI, weight %). LOI was detected after heating at 550 °C for 6 h and at 950 °C for 2 h [54]. The soil pH was defined potentiometrically using an ionometer, “Anoin-4100” (Novosibirsk, Russia), at a soil to water ratio of 1:25 for peat and 1:2.5 for mineral sediments, respectively.

#### 3.2.4. Definition of Amino Acid Composition

The quantitative content of AA composition was defined after hydrolysis of the samples with a HCl solution at a concentration of 6 mol/dm^3^, using liquid chromatography on ion exchange resins with the amino acid analyzer T-339 (Microtechna Praha, Prague, Czech Republic). AA mixtures were divided into individual components on a chromatographic column filled with the ion exchange resin, “Ostion” (Microtechna Praha, Prague, Czech Republic). The content of each AA in the eluate was determined photometrically by absorption at λ = 520 nm, and a yellow-colored compound was formed as a result of the post-column reaction of an AA with ninhydrin (according to the certified method of QCA MVI No. 88-17641-97-2010).

#### 3.2.5. Definition of Microbial Community Structure: Total Microbial DNA Extraction

DNA samples were extracted from a 0.5 g peat sample using a FastDNA™ Spin Kit for Soil (MP Biomedicals, Los Angeles, USA) reagent kit. The quantification of bacteria, archaea, and fungi communities in the studied microbiomes was evaluated using a quantitative polymerase chain reaction (qPCR) with real-time detection in a CFX96 Touch amplifier (BioRad, Moscow, Russia). Cloned fragments of the *Escherichia coli* (Sigma) ribosomal operon were used as a control for bacteria, *Halobacterium salinarum* strain FG-07 [55] was used for archaea, and the yeast strain *Saccharomyces cerevisiae* Meyen 1BD1606 [56] was used for fungi. The BioMaster HS-qPCR SYBR Blue(2×) mixture (Biolabmix, St-Petersburg, Russia) was applied for amplification. The following primers were used: Eub338/Eub518 for bacteria [57], arc915f/arc1059r for archaea [58], and ITS1f/5.8s for fungi [59]. Quantitative estimates were given in terms of the number of rRNA operons per gram of soil. Each sample was defined in triplicate. The qPCR results were processed by the software supplied with the CFX96 Touch amplifier (BioRad).

Based on the qPCR results, expressed as the number of copies of the rRNA operon per gram of soil, the bacterial, archaeal (number of cells), and fungal biomasses were estimated. This parameter is used to analyze the relative amounts of microorganisms in various soil; however, it also allows us to draw conclusions concerning the absolute number of a certain microorganism in a group.

## 4. Conclusions

The AA availability for uptake by microorganisms is impacted by a warming climate and it may affect the peatland carbon cycle response. Seventeen AAs have been identified and quantified. The share of AA fragments varies from 2.6 to 7.8, and it significantly highly correlates (r = 0.87, −0.74 and 0.67, *p* ˂ 0.05) with the soil’s organic nitrogen concentration, C/N ratio, and *δ*^15^N. Changes in the AA composition of organic matter throughout the peat strata depend on the changes in the plant macrofosssil composition of peat, which is manifested in the differences between the relative molar fractions of AA groups (acidic, basic, neutral, cyclic). The biological soil activity can be reflected by the molar fraction ratio of hydroxy to heterocyclic AAs, and this ratio can be a humification depth marker.

Our study shows clear differences between permafrost peatlands in terms of their microorganism distributions along their vertical structure. At sites Inta 1 and Inta 11, the number of microorganisms is at its maximum in the upper peat layers, and it sharply decreases as the depth increases. At sites Kolva and Voda-Ty, the microorganism distribution along the profile of peatlands has a wave pattern, with maximum values at a depth of 100–110cm and 190–210 cm in their respective permafrost layers. The high number of microorganisms is mainly determined by increased aeration, and hydrothermal conditions promoted the development and functioning of the microbial community in the upper peat layers at sites Inta 1 and Inta 11. The presence of large numbers of microorganisms in the deep peat layers, as revealed at sites Kolva and Voda-Ty, is not demonstrative of their high biological activity; at this depth (anaerobic conditions and negative temperatures due to permafrost), microorganisms are in an inactive state, and they form a microbial community, which determines the stability of microbial cenosis under changing external conditions. Bacteria are able to develop in microloci; therefore, even a low presence of oxygen in the peat strata is sufficient for bacterial reproduction throughout the profile. However, it is possible that a large number of microorganisms in the anaerobic zone of the catotelm is explained by the accumulation of facultative anaerobic forms, which can live both in aerobic and anaerobic conditions, passing from one type of metabolite to another.

The lower abundance of micromycetes, in comparison with bacteria and archaea. is a common characteristic for the microbial communities in the studied peatlands. The decrease in fungi content, which is correlated with the decrease in other microorganisms of the hydrolytic complex, is one of the reasons for the slow destruction of plant residues. Micromycetes are sensitive to the unfavorable conditions in the peat deposits, such as anaerobic conditions, the predominance of ultramicropores in the pore space, lower acidity, and soil invertebrates; therefore, their development as functionally active microorganisms is only possible in the upper layers of peat. The presence of fungi in the deeper layers is only possible in the form of spores.

Our research confirms the hypothesis that microorganisms in peatlands, regardless of their location, are present throughout the peat strata and are characterized by high numbers, even in deep peat layers. It is possible that most of the microorganisms under anaerobic conditions in the peat deposit are not functionally active; however, they nevertheless form a dormant pool. If there is a change in climatic conditions, and/or an anthropogenic disturbance in the upper biologically active peat layers, the pool of microorganisms will become active, and the homeostasis of the peatland as a system will be preserved. Moreover, processes such as the release of greenhouse gases can also accelerate.

The obtained data indicate that the control of the abundance and diversity of microorganisms at the ecosystem level differs in peat from well-drained soils. Under such conditions, it is not clear whether models of regularities with regard to the functioning of microbial communities in mineral soils in peatlands are applicable; thus, the regional trend regarding the microbial functional composition of peatland ecosystems deserves further attention in future studies.

There is no significant correlation between the microorganism distribution and plant macrofossil composition of the peat strata. The determining factors for the development of microorganism abundance are aeration and hydrothermal conditions.

Climate change will affect the carbon and nitrogen cycle in the peatlands. The obtained data analysis shows the existence of a large pool of microorganisms that is concentrated in permafrost peatlands and a vertical continuum of bacteria, archaea, and microscopic fungi along the peat profile, due to the presence of microorganisms in each layer, throughout the entire peat strata. Increasing temperatures and lowering groundwater levels will increase the microbial productivity of the peatlands, even in deeper layers; therefore, large reserves of carbon that are currently stored, will be released into the atmosphere. The limited availability of nitrogen will limit the ability of plants and microorganisms to respond to changing environmental conditions; however, with the increased decomposition of organic matter, amino acids will be released as organic sources of nitrogen as they are stored in the protein material of peat-forming plants and microbial communities, which can also affect the organic nitrogen cycle.

## Figures and Tables

**Figure 1 plants-11-02704-f001:**
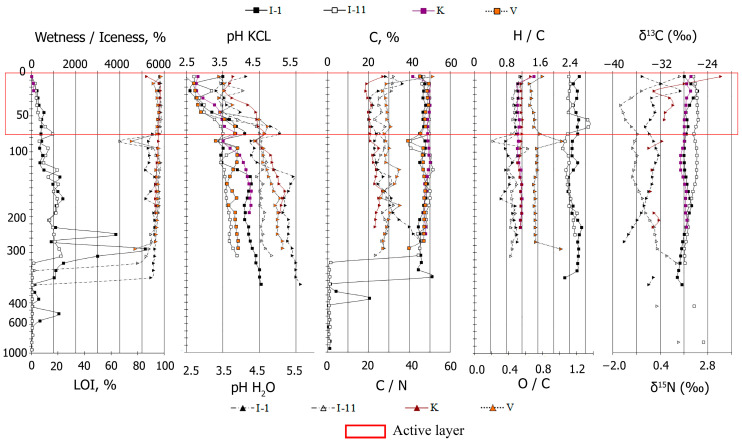
Physical and chemical properties of peat strata. LOI is loss of ignition.

**Figure 2 plants-11-02704-f002:**
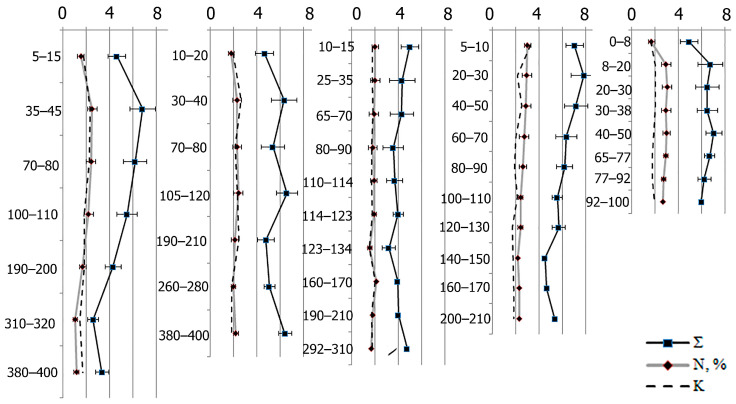
Profile change in terms of the sum of amino acids (Σ), the mass fraction of nitrogen in dry matter (N, %), and the mole fraction ratio of hydroxy AAs to heterocyclic AAs (K).

**Figure 3 plants-11-02704-f003:**
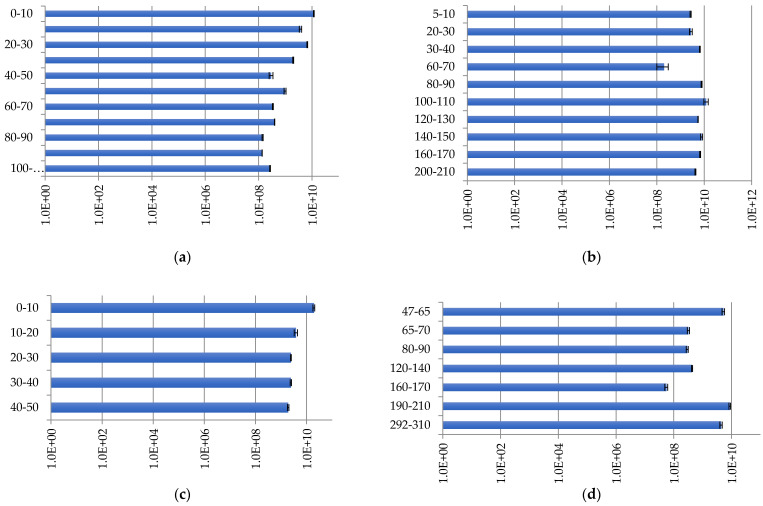
Bacterial distribution (copies/g of soil) in peat strata: (**a**) Inta 1; (**b**) Kolva; (**c**) Inta 11; and (**d**) Voda-Ty.

**Figure 4 plants-11-02704-f004:**
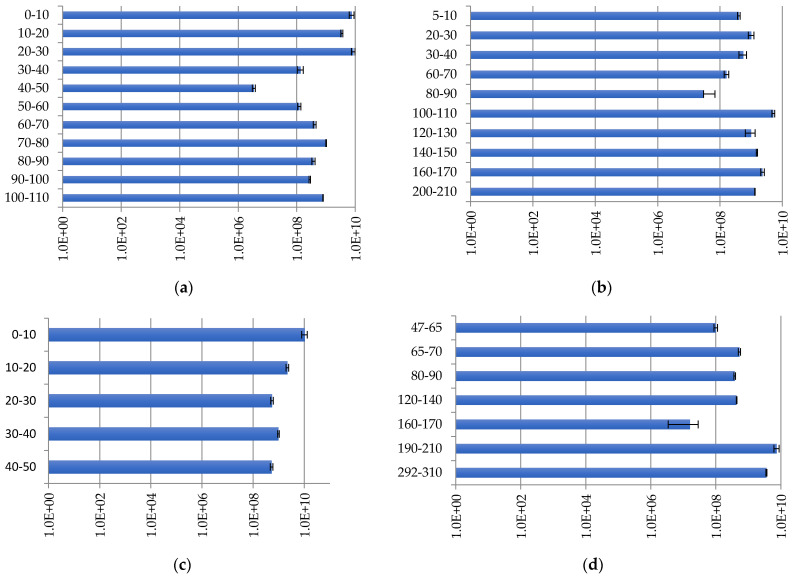
Archaeal distribution (copies/g of soil) in peat strata: (**a**) Inta 1; (**b**) Kolva; (**c**) Inta 11; and (**d**) Voda-Ty.

**Figure 5 plants-11-02704-f005:**
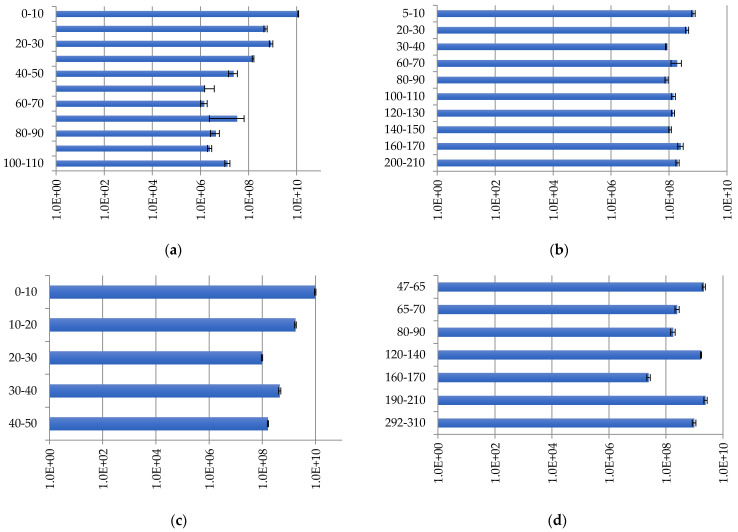
Fungal distribution (copies/g of soil) in peat strata: (**a**) Inta 1; (**b**) Kolva; (**c**) Inta 11; and (**d**) Voda-Ty.

**Figure 6 plants-11-02704-f006:**
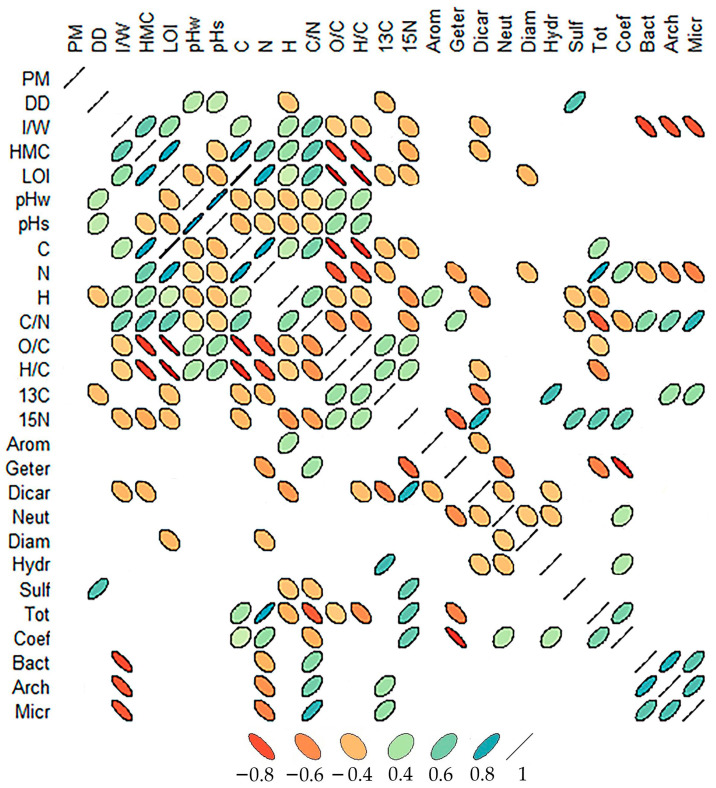
Pearson correlation coefficients between the amino acid composition, microbial community structure and the main physico-chemical peat properties. PM—plant macrofossils; DD—decomposition decree; %, I/W—Iciness/Wetness; HMC—Hygroscopic moisture coefficient; LOI—loss of ignition, %; pHw—pH H_2_O; pHs—pH K_2_O; C—soil’s organic carbon, %; N—soil organic nitrogen, %; H—total hydrogen, %; C/N—C/N ratio; O/C—O/C ratio; H/C—H/C ratio; 13C—*δ*^13^C, ‰; 15N—*δ*^15^N, ‰; Arom—Aromatic; Geter—Heterocyclic; Dicar—Dicarboxylic; Neut—Neutral, Diam—Diaminoacids; Hydr—Hydroxyacids; Sulf—Sulfur; Tot—Sum of amino acids; Coef—the mole fraction ratio of hydroxy AAs to heterocyclic AAs; Bact—Bacteria, copies per gram of peat; Arch—Archaea, copies per gram of peat; Micr—Micromycetes, copies per gram of peat.

**Figure 7 plants-11-02704-f007:**
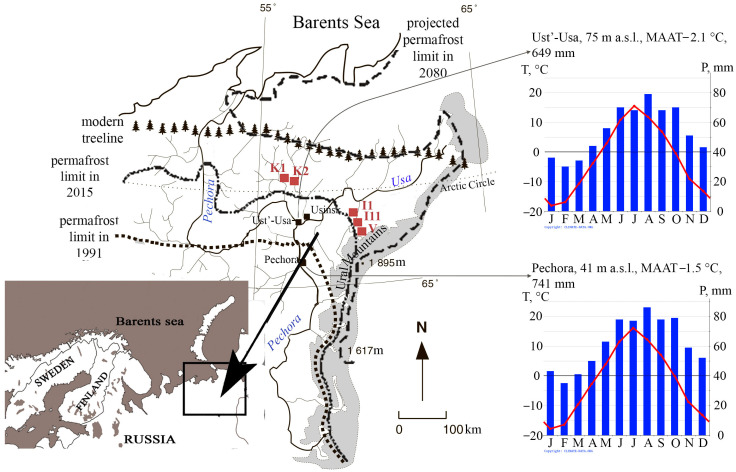
Key study sites: Inta 1, Inta 11, Kolva, and Voda-Ty (indicated by red squares). The closest climate stations are marked by black squares. Monthly air precipitation and temperatures were based on the climate database https://ru.climate-data.org/ (accessed on 31 August 2022) [47,48].

## Data Availability

Not applicable.

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
