# Peer review of "Microbial Community Structure in Ancient European Arctic Peatlands"

_plants, 2022, doi:10.3390/plants11202704_

Round 1
Reviewer 1 Report
Plants
Title: Differences in microbial community in ancient European Arctic peatlands
Author: A. Pastukhov
General Comments
The paper examines microbial community composition and some aspects of peat soil chemistry among four northern peatlands in Russia. The idea is not new but data on the subject are always interesting.
The data for amino acids is particularly interesting.
Specific Comments
1) In the Abstract, indicate the aim of the study, such as ‘to characterize microbial community composition along a gradient of climate.’
2) The Introduction is okay. However, consider adding a few sentences to the last paragraph that describe where the study was done.
3) The organization of the paper is a bit odd with introduction, results, methods, conclusion. Either put methods before results or after conclusion.
4) The text in the figures needs to be larger. They are difficulty to read.
5) Avoid one sentence paragraphs. A paragraph develops an idea with a topic sentence, argument, and conclusion. You cannot achieve this in one sentence.
6) Figure 6 is difficult to read. Consider showing only the correlations with values greater than 0.6. This indicates that the relationship explained, at least, 36% of the variation, which is still a bit low. Correlations with coefficients less than 0.6 are not particularly interesting.
7) The Methods are okay, to me. However, the sample sizes are not clear to me.
8) The Conclusion could be better. This is mostly a summary of the findings. Try to write a statement describing how the results provide a better understanding of peatlands in a changing climate.
Technical Comments
1) Title: delete ‘differences in.’
2) Line 9: consider saying, ‘northern peatlands are crucial reservoirs, etc.’
3) Line 10” consider saying, ‘and are vulnerable etc.’
4) Line 16: consider saying, ‘the data showed the existence, etc.’
5) Line 21: delete ‘limited.’
6) Line 29: delete ‘according to the latest estimates.’
7) Line 31: delete ‘participating in the global carbon and nitrogen cycle in the biosphere’ and say, ‘these systems provide global sinks, etc.’
8) Line 33: say ‘peatland systems act as powerful sources of, etc.’
9) Line 34: delete ‘will become.’
10) Line 37: delete ‘in northern bog ecosystems.’
11) Line 41: consider saying, ‘therefore a better understanding of the microbial community is needed.
12) Line 49: delete ‘nowadays.’
13) Line 49 to 60: avoid one sentence paragraphs. Combine into one paragraph. Start by saying, ‘we need better knowledge of microorganisms with depth in a given peatland and across different peatlands.’ After that explain why a better understanding is needed.
14) Line 61 to 72: try to combine as one paragraph. The ideas are good.
15) Line 88: move the sentence to line 82. It makes a good topic sentence for the past paragraph.
16) Line 91: delete ‘studied.’
17) Line 119: delete or explain in more detail.
18) Line 122: is this paragraph necessary? Start with the findings.
19) Line 128: spell ‘seventeen.’ Also say, seventeen different amino acids, etc.’
20) Line 139: delete the commas.
21) Line 158: change is based on’ to ‘can be described by.’
22) Line 185: is this paragraph necessary? Consider deleting it.
23) Line 222: again, is this paragraph necessary? Consider deleting it.
24) Line 235 -245: combine paragraphs.
25) Line 247 – 252: is this part necessary? Consider deleting it.
Author Response
Response to Review Report Form
Dear reviewer,
Many thanks for your great job and very valuable comments and remarks improving our manuscript. We also edited of English and style of our manuscript.
Title: Differences in microbial community in ancient European Arctic peatlands
Author: A. Pastukhov
General Comments
The paper examines microbial community composition and some aspects of peat soil chemistry among four northern peatlands in Russia. The idea is not new but data on the subject are always interesting.
The data for amino acids is particularly interesting.
Specific Comments
1) In the Abstract, indicate the aim of the study, such as ‘to characterize microbial community composition along a gradient of climate.’
The aim was to characterize the amino acids and microbial community composition in peat strata along a gradient of climate. The amino acids and microbiota diversity were studied by liquid chromatography and a quantitative polymerase chain reaction.
2) The Introduction is okay. However, consider adding a few sentences to the last paragraph that describe where the study was done.
The study aim is to fill some of these knowledge the gaps by having determined the structural features and given an assessment of the profile distribution of microorganisms and found correlations with the amino acid content in northern permafrost peatlands at the southern cryolithozone limit of the cryolithozone.
Lines 336-349: The detailed description of study sites.
3) The organization of the paper is a bit odd with introduction, results, methods, conclusion. Either put methods before results or after conclusion.
We used instructions for authors to prepare the manuscript (Microsoft Word template)
4) The text in the figures needs to be larger. They are difficulty to read.
Corrected
5) Avoid one sentence paragraphs. A paragraph develops an idea with a topic sentence, argument, and conclusion. You cannot achieve this in one sentence.
Corrected
6) Figure 6 is difficult to read. Consider showing only the correlations with values greater than 0.6. This indicates that the relationship explained, at least, 36% of the variation, which is still a bit low. Correlations with coefficients less than 0.6 are not particularly interesting.
Corrected
7) The Methods are okay, to me. However, the sample sizes are not clear to me.
The diameter of measuring cylinder and drilling machine pipe is 100 mm.
8) The Conclusion could be better. This is mostly a summary of the findings. Try to write a statement describing how the results provide a better understanding of peatlands in a changing climate.
Corrected
Technical Comments
1) Title: delete ‘differences in.’
Ok
2) Line 9: consider saying, ‘northern peatlands are crucial reservoirs, etc.’
Corrected
3) Line 10” consider saying, ‘and are vulnerable etc.’
Corrected
4) Line 16: consider saying, ‘the data showed the existence, etc.’
Corrected
5) Line 21: delete ‘limited.’
Ok
6) Line 29: delete ‘according to the latest estimates.’
Ok
7) Line 31: delete ‘participating in the global carbon and nitrogen cycle in the biosphere’ and say, ‘these systems provide global sinks, etc.’
Corrected
8) Line 33: say ‘peatland systems act as powerful sources of, etc.’
Corrected
9) Line 34: delete ‘will become.’
Ok
10) Line 37: delete ‘in northern bog ecosystems.’
Ok
11) Line 41: consider saying, ‘therefore a better understanding of the microbial community is needed.
Corrected
12) Line 49: delete ‘nowadays.’
Ok
13) Line 49 to 60: avoid one sentence paragraphs. Combine into one paragraph. Start by saying, ‘we need better knowledge of microorganisms with depth in a given peatland and across different peatlands.’ After that explain why a better understanding is needed.
Ok
14) Line 61 to 72: try to combine as one paragraph. The ideas are good.
Ok
15) Line 88: move the sentence to line 82. It makes a good topic sentence for the past paragraph.
Ok
16) Line 91: delete ‘studied.’
Ok
17) Line 119: delete or explain in more detail.
Deleted
18) Line 122: is this paragraph necessary? Start with the findings.
Changed
19) Line 128: spell ‘seventeen.’ Also say, seventeen different amino acids, etc.’
Corrected
20) Line 139: delete the commas.
Deleted
21) Line 158: change is based on’ to ‘can be described by.’
Changed
22) Line 185: is this paragraph necessary? Consider deleting it.
We suggest, yes
23) Line 222: again, is this paragraph necessary? Consider deleting it.
Yes, it is necessary.
24) Line 235 -245: combine paragraphs.
Combined
25) Line 247 – 252: is this part necessary? Consider deleting it.
Yes, it is necessary.

Reviewer 2 Report
- This article studies the microbial communities in peatlands soil and carries out correlation analyses for potential nutrient cycling consequences. The writing is clear. Several questions need to be answered.
- Since the background information has broadly described the challenges and significance for peatlands, it will be useful to draw some implications of how the problems can be potentially relieved.
- The topic can be more precise to include information about the aims/ content.
- A conceptual diagram can be given to describe the potential mechanisms.
- Regarding microbial niches for anaerobic/anoxic zones (due to mass transfer), one reference about granular biomass (having confocal imaging) is suggested. Quantitative characterization and analysis of granule transformations: Role of intermittent gas sparging in a super high-rate anaerobic system
Author Response
Response to Review Report Form
Dear reviewer,
Many thanks for your great job and very valuable comments and remarks improving our manuscript. We also edited of English and style of our manuscript. We added the necessary information to the text of the article. It is highlighted in blue.
- This article studies the microbial communities in peatlands soil and carries out correlation analyses for potential nutrient cycling consequences. The writing is clear. Several questions need to be answered.
- Since the background information has broadly described the challenges and significance for peatlands, it will be useful to draw some implications of how the problems can be potentially relieved.
- The topic can be more precise to include information about the aims/ content.
- A conceptual diagram can be given to describe the potential mechanisms.
- Regarding microbial niches for anaerobic/anoxic zones (due to mass transfer), one reference about granular biomass (having confocal imaging) is suggested. Quantitative characterization and analysis of granule transformations: Role of intermittent gas sparging in a super high-rate anaerobic system
- Thank you for your interest in the problems of our study and for your valuable comments, which we tried to take into account in the editorial of the paper.
- In the "Discussion", findings are formulated and added to expand the understanding of the solution of the tasks set in the study. Our findings are added to the “Conclusion” on the need to create a new model for monitoring the abundance and diversity of microorganisms in peatlands, since existing approaches to the study of microbial communities are applicable only to mineral soils.
- From the point of view of all authors, the paper topic fully reflects the aim and objectives of the study.
- Our studies of the functioning of microbial communities in permafrost peatlands of the Arctic sector are the initial stage, which currently does not allow us to apply the conceptual scheme for the studied mechanisms. However, this is possible in our further studies.
- Under the conditions of this study, no tasks were set to determine the role of different microorganism groups, including anaerobic ones, in the processes of transformation of organic matter and the release of greenhouse gases. The problem of the functioning of microorganisms in permafrost peatlands has not been fully studied, which allows us to solve these problems in the future.
